# Use of Long-Acting Injectable Antipsychotics in Inpatients with Schizophrenia Spectrum Disorder in an Academic Psychiatric Hospital in Switzerland

**DOI:** 10.3390/jpm12030441

**Published:** 2022-03-11

**Authors:** Stephan Reymann, Georgios Schoretsanitis, Stephan T. Egger, Alexey Mohonko, Matthias Kirschner, Stefan Vetter, Philipp Homan, Erich Seifritz, Achim Burrer

**Affiliations:** 1Department of Psychiatry, Psychotherapy and Psychosomatics, Psychiatric Hospital, University of Zurich, 8032 Zurich, Switzerland; stephan.reymann@pukzh.ch (S.R.); georgios.schoretsanitis@pukzh.ch (G.S.); stephan.egger@pukzh.ch (S.T.E.); alexey.mohonko@pukzh.ch (A.M.); matthias.kirschner@pukzh.ch (M.K.); stefan.vetter@pukzh.ch (S.V.); philipp.homan@pukzh.ch (P.H.); erich.seifritz@bli.uzh.ch (E.S.); 2Neuroscience Center Zurich, University of Zurich, 8057 Zurich, Switzerland

**Keywords:** long-acting injectable, antipsychotic, depot, schizophrenia spectrum disorder, schizophrenia, schizoaffective, inpatient, prescribing pattern

## Abstract

Long-acting injectable antipsychotics (LAIs) offer many benefits to patients with schizophrenia spectrum disorder (SSD). They are used with very different frequencies due to questions of eligibility or patients and prescribers’ attitudes towards LAI use. We assessed the prescribing rates of LAIs in a large academic psychiatric hospital with a public service mandate in Switzerland and compared them with other countries and health care systems. To our knowledge, this study is the first to investigate inpatient LAI use in Europe. Medical records of all patients diagnosed with SSD discharged from the Clinic of Adult Psychiatry of the University Hospital of Psychiatry Zurich over a 12 month period from January to December 2019 were evaluated regarding the prescribed antipsychotics at the time of discharge. The rates of use of LAIs among all patients and among patients receiving LAI-eligible antipsychotic substances were assessed retrospectively. We assessed records of 885 patients with SSD. Among all cases, 13.9% received an LAI. Among patients who received antipsychotic medication that was eligible for LAI use (*n* = 434), 28.1% received an agent as an LAI. LAI use included paliperidone palmitate (69.9%), aripiprazole monohydrate (14.6%), risperidone (4.9%) and first-generation LAIs (9.8%). Compared to international frequencies of LAI administration, the prescription rate of LAIs in SSD patients was low. Further studies will evaluate patient- and prescriber-related reasons for this low rate.

## 1. Introduction

Long-acting injectable antipsychotics (LAIs) emerged as an important and effective treatment option for patients with schizophrenia spectrum disorder (SSD), including schizophrenia, and are particularly effective in reducing hospitalization [1]. In the US, among all patients receiving an antipsychotic agent eligible for LAI use, 4–28% receive their agent as an LAI [2,3,4].

LAIs were initially developed to improve adherence in patients suffering from SSD. However, recent clinical studies suggest additional advantages [5]. First, the prescription of LAIs is associated with better adherence in a randomized controlled trial and a retrospective analysis [6,7]. For instance, a recent meta-analysis of prospective and retrospective cohort studies found that the risk for all-cause discontinuation was lower in patients receiving LAIs than in patients receiving oral antipsychotics (OAPs) [1]. Second, patients treated with LAIs are at lower relapse risk compared to patients receiving OAPs in a meta-analysis of randomized trials [8]. Third, LAIs provide prolonged sustainable antipsychotic effects even if discontinued, as shown in a recent re-analysis of five placebo-controlled randomized trials, where the time to relapse after discontinuation of medication was significantly longer in patients receiving long-acting paliperidone palmitate than in those receiving paliperidone orally before discontinuation [9]. Fourth, LAIs significantly reduce the rate of hospitalization, according to a recent meta-analysis [10]. Thus, LAIs are not only clinically effective but also reduce medical costs, as shown in a meta-analysis [11]. Another benefit of LAIs is the lower risk of being arrested or incarcerated in patients treated with LAIs compared to OAPs, in a randomized review board-blinded study [12]. Finally, evidence of a meta-analysis of randomized controlled studies suggests similar tolerability of LAIs and OAPs [13]. Possible disadvantages of LAIs include the fact that treatment cannot be stopped immediately in the case of side effects or clinical indications of a switch to another agent, as well as the common risks and side effects of intramuscular injections. In addition, in the case of withdrawal of patients’ acceptance, the treatment cannot be discontinued immediately. However, a recent meta-analysis by Yaegashi et al. suggested that there was no significant difference in cessation of treatment due to withdrawal of consent, comparing patients treated with LAIs to those treated with oral antipsychotics [14]. Concerning side effects, LAIs were associated with higher rates of akinesia, low density lipoprotein change and anxiety but showed a lower rate of prolactin change in a meta-analysis of randomized controlled trials comparing LAIs and OAPs [13]. 

Currently, four second-generation antipsychotics are available as LAI formulations: risperidone, paliperidone, aripiprazole and olanzapine [15]. Availability varies between countries, e.g., the olanzapine LAI is not approved in some European countries, including Switzerland. Among the first-generation antipsychotics, haloperidol, zuclopenthixol, fluphenazine, perphenazine and flupenthixol are available as LAI formulations [16].

In light of the current evidence, LAIs seem a valuable treatment option for patients with SSD. Yet, data regarding prescription patterns of LAIs and related temporal trends are required. In a large study with data from the French healthcare system, it could be shown that the proportion of patients receiving LAIs among those receiving any antipsychotic medication increased over the years 2007 to 2014 from 8.9 to 9.6%, regardless of diagnosis or treatment setting [17]. However, there is a lack of evidence about the frequency of LAI prescription in an inpatient setting. Moreover, available data from the United States present substantial variation; for example, a study from 2015 suggested that only 9% of inpatients were prescribed LAIs within the first 30 days after a schizophrenia-related hospitalization, using data from Medicaid programs from multiple US states [7]. Similar trends were obtained by Kishimoto et al. in an inpatient setting in New York City [18]. On the other hand, in a hospital mainly serving a low-income and uninsured population in Brooklyn, New York City, the authors reported that 44% of inpatients were treated with LAIs at the time of discharge [19]. In addition to these three studies, there are, to our knowledge, no other data evaluating the use of LAIs in acute psychiatric inpatient settings; additionally, none of these studies analyzed LAI prescription rates in Europe. In a randomized clinical intervention trial, it was shown that LAI-focused staff training potentially enhances the use of LAIs (91.0% in the intervention group vs. 51% in clinician’s choice treatment) [20,21].

The aim of this descriptive retrospective study was to assess prescription rates of LAIs in a large academic psychiatric hospital in Switzerland. The University Hospital of Psychiatry Zurich is a public hospital with a service mandate for psychiatric care in a mixed urban and rural region covering approximately 500,000 inhabitants. Part of the service mandate is to provide treatments for all acute and chronic psychiatric disorders. As part of the Hospital, the Clinic of Adult Psychiatry (Department of Psychiatry, Psychotherapy and Psychosomatics) offers psychiatric inpatient units for acute admissions. 

## 2. Materials and Methods

We conducted a retrospective analysis of inpatients with SSD treated in the University Hospital of Psychiatry Zurich and assessed the rates of LAI use in this group. As part of the clinical routine, diagnoses were provided using ICD-10 criteria. All patients with a WHO-ICD-10 Chapter F2 SSD were included: schizophrenia (F20), schizotypal disorder (F21), delusional disorder (F22), brief psychotic episode (F23) and schizoaffective disorder (F25). All patients with SSD receiving antipsychotic substances that are available for LAI use in Switzerland were included, regardless of prior hospitalizations, time since onset of the disease or possible contraindications.

Data from all patients diagnosed with SSD discharged from the Clinic of Adult Psychiatry of the University Hospital of Psychiatry Zurich over a 12 month period from January until December 2019 were included. Patients’ data were retrospectively reviewed, and psychiatric diagnoses and prescribed antipsychotic medication at time of discharge from hospital were assessed. 

In the prescribed discharge medication, antipsychotics were evaluated in terms of their availability as LAI formulations approved in Switzerland and thereby classified as “eligible” or “not eligible”. The eligible substances were risperidone, paliperidone, aripiprazole, haloperidol, zuclopenthixol and flupenthixol. We rated whether the substances, if eligible, were prescribed orally or administered as LAI formulations. 

The project was approved by the Ethics Committee of the Canton of Zurich (BASEC-Nr. Req-2021-00376). 

Statistical analyses were carried out using SPSS 27 (IBM). We performed a descriptive analysis of the sample regarding age, sex, diagnostic subgroups, comorbid substance use disorders, length of stay and LAI prescription rate and type. Frequencies and proportions of the above-mentioned demographic and clinical data were calculated. 

Among all patients, the two groups of patients receiving LAIs vs. patients receiving oral antipsychotics were compared with regard to age and length of stay using a two-sample *t*-test and concerning the categorical variables gender and presence of substance use disorder using Chi square test (X^2^).

## 3. Results

A total of 2203 records of inpatients were screened, of which 40.2% (*n* = 885) were diagnosed with SSD and enrolled in the study; 62.9% (*n* = 557) of all patients diagnosed with SSD were male, and 37.1% (*n* = 328) were female. The mean age was 40.51 years (SD 12.44). Of the 885 patients enrolled, 89.7% (*n* = 794) were prescribed an antipsychotic medication at discharge. Of these, 49.0% of patients (*n* = 434) were treated with an antipsychotic agent that is available for both oral and LAI administration (classified as “eligible” for LAI use). Among those 434 patients, 28.1% (*n* = 122) received LAI antipsychotics. Additionally, one patient (*n* = 1) received olanzapine LAI as off-label. In total, 13.9% (*n* = 123) of all SSD cases received an LAI. Among the group of patients receiving an LAI, 69.9% (*n* = 86) were treated with paliperidone palmitate, 14.6% (*n* = 18) with aripiprazole monohydrate, 4.9% (*n* = 6) with risperidone microspheres and 9.8% (*n* = 12) with a first-generation LAI. One patient (0.8%) received olanzapine pamoate as off-label, as it is not approved in Switzerland. The characteristics of the patients and of the diagnoses and the prescribed medications in the respective diagnostic subgroups are summarized in Table 1. The highest rates of LAI prescription were found in patients diagnosed with a schizoaffective disorder (16.0%, *n* = 25), followed by the patients diagnosed with schizophrenia (15.5%, *n* = 91), who represented the largest subgroup in our sample (66.2% of all patients, *n* = 586). In the other subgroups, LAIs were prescribed less often, and none of the five patients with a schizotypal disorder received an LAI. The patients receiving LAIs were significantly younger compared to non-LAI-treated patients (36.87 SD 11.53 vs. 41.09 SD 12.49 years, *p* < 0.01), and there were more males among the LAI-treated patients (74.0%, *n* = 91 vs. 61.2%, *n* = 466, *p* = 0.006) (Table 2). Furthermore, among patients with LAIs, there were significantly more patients diagnosed with any comorbid substance-use disorders compared to patients without LAIs (56.9% vs. 33.9%, *p* < 0.01). Among the 123 cases receiving an LAI, in 57% (*n* = 70), the LAI was prescribed prior to hospitalization, whereas in 43% (*n* = 53) of patients, the LAI was newly started during the current hospitalization. Patients with LAIs newly started compared to patients with no LAI had a mean age of 36.51 vs. 41.09 years (*p* = 0.01), a length of stay of 37.28 vs. 24.08 days (*p* < 0.001), a number of hospitalizations of 9.55 vs. 9.91 (*p* = 0.859) and a time since first admission of 7.93 vs. 8.60 years (*p* = 0.559). Patients with LAIs newly started compared to patients without LAIs were more frequently male (67.9% vs. 61.1%; *p* = 0.011), and a substance-use disorder was more frequent (39.6% vs. 33.9%; *p* < 0.001). 

## 4. Discussion

In our sample, the prescription rate for LAI antipsychotics in patients with SSD is lower than reported elsewhere. Patients receiving LAIs are predominantly male and younger than those who are not prescribed an LAI. 

Of the patients treated with an antipsychotic substance that is available as an LAI, 28.1% were treated with an LAI. While this rate seems rather promising, one has to consider that only 13.9% of all patients with SSD were treated with an LAI, which appears rather low. The LAI prescribed most often was by far paliperidone palmitate, followed by aripiprazole monohydrate; risperidone microspheres and first-generation LAIs were prescribed in a small patient subgroup. These results are in contrast with the findings of Olayinka et al. [19], who assessed LAI prescription rates in an inpatient setting and reported that 44% of a total of 43 patients with SSD were treated with an LAI. They found risperidone microspheres to be the most often used substance, followed by paliperidone palmitate and haloperidol decanoate. In a study by Kishimoto et al., in an inpatient setting [18], 32.9% of the patients were found to be discharged on an LAI. Similar results were found in another large study in 2009 by Barnes et al. [22], where 35% of 2032 acute inpatients and 28% of a forensic patient sample were found to be treated with an LAI. As a limitation, one has to note that, at that time, most commonly, first-generation antipsychotic LAIs were used, and risperidone microspheres was the only available second-generation antipsychotic LAI. A potential reason underlying the differences regarding the LAI prescription rates estimated in our sample and other studies, such as, for example, the study of Olayinka et al. in Brooklyn, New York, [19], may be associated with differences in the socio-economic status of the patients included and the healthcare system. In fact, there is also a considerable difference in the duration of hospitalization (24.62 days average length of stay in our study vs. 14 days in the study by Olayinka et al. [19]). In Switzerland, the longer lengths of stay could be used for further discussion of the advantages and disadvantages of LAIs. A possible reason for the higher LAI prescription rates in Brooklyn might be leaning more to a recommendation of LAI use. In relation to counselling on LAI use, it is well known that the prescriber and patient’s attitudes in terms of LAIs might influence the rate of LAIs and that stigmatization and prejudices involving LAIs can be reduced by well-informed shared decision making [23,24,25,26]. Common negative factors are the prescriber’s attitude that LAIs might not be suitable for the treatment of first psychosis [24], while the opposite can be proven [20]. Another point is the fact that the selected agent is not eligible for LAI use [24]. The second point might partly explain the apparently low rate of LAI use in our study. In Switzerland, olanzapine, which is in our sample the most used antipsychotic substance (29.5% of all patients with antipsychotic medication), is not available for LAI use, while it is largely available in other parts of the world, including the US and UK. If olanzapine pamoate was available in Switzerland in the study, both the rate of patients treated with an eligible substance for LAI and probably also the rate of LAI use among all patients might have been higher. Other interesting findings in our study were that there were not only more male patients in the sample than female, but also there were significantly more male patients receiving LAIs than female patients—findings that are in line with other studies [22,27]. Furthermore, patients receiving LAIs in our study were significantly younger compared to those not receiving LAIs, indicating perhaps that in the recent years, offering LAIs to patients has become more frequent as new substances have come to market, whereas older patients either aren’t offered LAIs in the course of their illness or tend to refuse LAI use. They were also younger than seen in other studies [19,22].

Of the LAIs, 43% were newly installed during the assessed hospitalization. Since it can be assumed that a relevant part of the pre-installed 57% of LAIs were also installed during a prior hospitalization, inpatient setting seems to be an important factor of LAI introduction in Switzerland. While inpatient setting allows LAI installation during an acute phase of treatment, reducing rates of relapse and rehospitalization, early LAI introduction in an outpatient setting might even prevent any hospitalization [10] and should therefore be offered as early as possible.

It is also of particular importance that patients with comorbid substance-use disorder more frequently received an LAI, as this group of patients is less adherent to therapy due to substance use [28].

As seen by the many advantages following LAI medication, increasing the rate of LAI use seems to be desirable. Therefore, measures to increase the rate should be evaluated [29,30]. First, it seems useful to inform each patient who is treated with an eligible substance about LAI availability, including the many advantages and the convenience of the LAI formulation, thereby motivating patients to consider LAI use. It is to be assumed that this does not take place equally everywhere, as not only patients but also clinicians’ attitude towards LAIs is often burdened with prejudices, as mentioned above. As shown in a recent study, a large number of early-phase schizophrenia patients accept therapy with LAIs after being offered these and informed following a standardized study protocol [21]. 

In order to bring these findings into clinical practice, it must be investigated whether the rate of LAI use could be increased by providing patient information and education in a standardized form, such as a brief information sheet or standardized interview, as well as by training clinicians on the existing evidence, to increase usage of LAIs [20,21,31]. Second, from our point of view, the observed low rates of LAI prescription in the present sample may further stimulate the ongoing discussion of whether the availability of a substance as an LAI formulation should be a selection criterion for the initial choice of an antipsychotic substance at the beginning of a treatment [20,32,33,34]. Third, it would be crucial to develop new agents in LAI formulations but also to increase efforts to develop LAI formulations for currently established compounds.

We state the following limitations: We provided numbers on patients being prescribed LAIs prior to hospitalization and patients newly started with LAIs during current hospitalization. However, we acknowledge that in patients receiving LAIs before hospitalization, we did not have data on whether medication with LAI was initiated in an outpatient setting or during a previous hospitalization. Therefore, the precise number of patients for whom an LAI was newly installed in an in- or outpatient setting cannot be provided. Furthermore, we state that standardized data on the duration of illness were not available. However, we considered providing a potential surrogate of illness duration, estimating the time since first admission. 

## 5. Conclusions

Only 13.9% of SSD inpatients received an LAI, as opposed to 86.1% who did not. This rate is low in light of the evidence for improved relapse prevention. This can be partly attributed to the non-availability of LAI formulations of some agents. However, among patients receiving an agent eligible for LAI use, still only 28.1% of the patients received an LAI. The underlying reasons—patient or prescriber related—remain to be further evaluated. Further research is needed to evaluate if this rate might be increased by standardized information procedures or motivational interventions.

## Figures and Tables

**Table 1 jpm-12-00441-t001:** Demographic and clinical data.

	All SSD 100% (*n* = 885)	Schizophrenia 66.2% (*n* = 586)	Schizoaffective Disorder 17.6% (*n* = 156)	Brief Psychotic Episode 13.2% (*n* = 117)	Delusional Disorder 2.4% (*n* = 21)	Schizotypal Disorder 0.6% (*n* = 5)
Age, years (mean, SD)						
	40.51 (12.44)	40.21 (12.27)	44.25 (11.78)	35.54 (11.83)	50.10 (12.14)	34.60 (14.54)

Gender, m/f (%, n)	m: 62.9% (*n* = 557)	m: 67.2% (*n* = 394)	m: 55.8% (*n* = 87)	m: 54.7% (*n* = 64)	m: 42.9% (*n* = 9)	m: 60.0% (*n* = 3)
	f: 37.1% (*n* = 328)	f: 32.8% (*n* = 192)	f: 44.2% (*n* = 69)	f: 45.3% (*n* = 53)	f: 57.1% (*n* = 12)	f: 40.0% (*n* = 2)
Number of hospitalizations (n, mean, SD)						
	10.6 (15.25)	11.5 (15.77)	15.2 (17.08)	1.9 (2.03)	3.5 (4.14)	3.8 (5.22)

Time since first admission, years (mean, SD)						
	8.7 (8.05)	9.3 (7.86)	12.3 (8.22)	2.0 (3.71)	3.5 (5.58)	5.3 (7.55)

Antipsychotic medication (%, n)						
	89.7% (*n* = 794)	92.7% (*n* = 543)	91.7% (*n* = 143)	80.3% (*n* = 94)	52.4% (*n* = 11)	60.0% (*n* = 3)

Medication eligible for LAI use (%, n)						
	49.0% (*n* = 434)	50.7% (*n* = 297)	47.4% (*n* = 74)	44.4% (*n* = 52)	42.9% (*n* = 9)	40% (*n* = 2)

LAI (%, n)	13.9% (*n* = 123)	15.5% (*n* = 91)	16.0% (*n* = 25)	5.1% (*n* = 6)	4.8% (*n* = 1)	0.0% (*n* = 0)
Paliperidone (% of LAI, n)	69.9% (*n* = 86)	64.8% (*n* = 59)	88% (*n* = 22)	83.3% (*n* = 5)		
Aripiprazole (% of LAI, n)	14.6% (*n* = 18)	14.3% (*n* = 13)	12% (*n* = 3)	16.7% (*n* = 1)	100% (*n* = 1)	
Risperidone (% of LAI, n)	4.9% (*n* = 6)	6.6% (*n* = 6)				
First-generation antipsychotic (% of LAI, n)	9.8% (12)	13.2% (*n* = 12)				
Olanzapine (% of LAI, n)	0.8% (1)	1.1% (*n* = 1)				

Demographic and clinical data of the total sample and diagnostic subgroups are shown: f: females; m: males; LAI: long-acting injectable; SD: standard deviation; SSD: schizophrenia spectrum disorder; Mean Age in years, percentage (%) and total number (n) of men and women, percentage (%) and total numbers (n) of fractions of antipsychotic medication as well as LAI eligible antipsychotic medication. Below, percentage (%) and total number (n) of patients receiving an LAI are shown. Among patients receiving an LAI, distribution of used LAI agents in total sample and diagnostic subgroups is shown in percentage (%) and total number of patients (n).

**Table 2 jpm-12-00441-t002:** Group comparisons.

	LAI (*n* = 123)	Non-LAI (*n* = 762)	Statistical Test	*p*-Value
Age, years (mean, SD)				
	36.87 (11.53)	41.09 (12.49)	*t* = 3.516	<0.01

Gender, m/f (%, n)	m: 74.0% (91)	m: 61.2% (*n* = 466)	χ^2^ = 7.472	0.006
	f: 26.0% (32)	f: 38.8% (*n* = 296)		
Substance use disorder (%, n)				
	56.9% (*n* = 70)	33.9% (*n* = 258)	χ^2^ = 24.13	<0.01

Length of stay, days (mean, SD)				
	27.95 (33.78)	24.08 (25.89)	*t* = −1.47	0.142

Number of hospitalizations (n, mean, SD)				
	15.12 (19.25)	9.91 (14.39)	*t* = −2.876	0.005

Time since first admission, years (mean, SD)				
	9.39 (7.63)	8.60 (8.11)	*t* = 1.010	0.313


Group comparisons: Statistical tests were performed to compare patients receiving an LAI with those not receiving an LAI. f: females; m: males; LAI: long-acting injectable; SD: standard deviation. Two-sample t-tests were performed for interval-scaled variables; Pearson Chi square tests were performed for categorical data. Mean age in years (a) and standard deviation (SD), percentage (%) and total number (n) of men and women, percentage (%) and total number (n) of fractions of antipsychotic medication as well as LAI-eligible antipsychotic medication.

## Data Availability

The datasets generated and analyzed during the current study are available from the corresponding author on reasonable request.

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
