# Peer review of "Use of Long-Acting Injectable Antipsychotics in Inpatients with Schizophrenia Spectrum Disorder in an Academic Psychiatric Hospital in Switzerland"

_jpm, 2022, doi:10.3390/jpm12030441_

Round 1

Reviewer 1 Report

This retrospective study examined the rate of LAI use in inpatients with schizophrenia spectrum disorders in Switzerland. The topic is timely and clinically important, the methods are straightforward, the results have significant implications for clinical practice and further research, and the manuscript is well-written.

I hope my comments will further improve the quality of the manuscript.

  1. LAIs can be introduced in both outpatient and inpatient settings. Which setting is more common is relevant to the frequency of LAI use in inpatients. I suggest that the authors report the number of inpatients who received LAI introduction (ie, newly started) during the hospitalization and compare characteristics between these patients and those not receiving LAIs.
  2. Related to the point above, the authors should discuss the merits and demerits of LAI introduction between outpatient and inpatient settings.
  3. The methods noted “… all patients with SSDs receiving antipsychotic substances that are available for LAI use in Switzerland were included…”. However, it seems that the authors included all patients with SSD receiving antipsychotics (n=885), regardless of antipsychotics available for LAI in Switzerland. Please clarify this.
  4. Please add the information on the duration of illness and the number of prior hospitalizations to Table 1 and 2 to illustrate the patient characteristics in more detail.
  5. In the context of patient acceptance of LAIs, the authors may want to cite this meta-analysis (https://pubmed.ncbi.nlm.nih.gov/33309187/) showing no difference in consent withdrawal in randomized controlled trials between LAIs and oral antipsychotics.

Reviewer 2 Report

The authors presented an interesting study on use of long-acting injectable antipsychotics in inpatients with schizophrenia spectrum disorder in an academic psychiatric hospital in Switzerland. It was descriptive retrospective study with the aim to assess prescription rates of LAIs in a large academic psychiatric hospital in Switzerland. The manuscript is well written, and the obtained statistic data well interpreted. The discussion is in-depth and addresses the many possible aspects contributing to a low LAIs prescription rate the academic psychiatric hospital in Switzerland. 

Minor comments: 

  1. Please add letter abbreviations (f and m) in Table 1 before the obtained data in a percentage (in the Gender row). As shown in table 2. 
  2. Please expand the FGA abbreviation under Table 1. 
  3. In lines 167, 170, 172 etc.: please add a period after the abbreviation et al. (There is an et al and it should be: et al.) 

Additionally, the authors might consider replacing Table 2 with a graph. 
